# Crisis, What Crisis? The Effect of Economic Crises on Spending on Online and Offline Gambling in Spain: Implications for Preventing Gambling Disorder

**DOI:** 10.3390/ijerph20042909

**Published:** 2023-02-07

**Authors:** Mariano Chóliz

**Affiliations:** Gambling and Technological Addictions Research Unit, Psychology School, University of Valencia, 46010 Valencia, Spain; mariano.choliz@uv.es

**Keywords:** gambling disorder, economic crises, online gambling, gambling disorder prevention

## Abstract

(1) Background: In the period between 2008 and 2020, the world experienced two global economic crises that affected people’s way of life and well-being: the financial crisis of 2008 and that precipitated by the COVID-19 pandemic. Despite the crises’ radically different causes, their consequences for economic activity have been equally dramatic; (2) Methods: This article analyzes the consequences of both crises on gambling spending in Spain and compares traditional (offline) games with more recent online offerings. The data were collected from databases maintained by the Spanish government and gambling companies; (3) Results: The paper offers two main conclusions. The first is that, while traditional (offline) gambling has been significantly affected by economic crises, online gambling has exhibited consistent growth since its legalization. The second is that the measures implemented to resolve the two economic crises differed significantly and thus had different impacts on spending on the various types of gambling; (4) Conclusions: The key conclusion is that purchasing power (measured in terms of GDP) can only explain spending on less addictive gambling games, such as lotteries. However, the availability and accessibility of games are both directly related to spending on games of all types.

## 1. Introduction

Gambling has existed since the earliest periods of civilization [1]. People have always staked money or material goods against other players with the expectation of deriving a profit, but with the risk of losing the bet. These characteristics of gambling differentiate it from other gaming activities pursued exclusively for enjoyment, or to promote the development of the personality or learning [2,3].

Gambling features in most cultures, and historical records of gambling date back to antiquity [1]. However, throughout history, gambling has been an essentially private activity, in which players derived their winnings from other gamblers’ losses. In principle, all players were equally likely to win and the losing player’s assets were transferred in their entirety to the winner. The end of the early modern period witnessed a substantial change in gambling practices. What had hitherto been a private activity between individual players became a form of tax collection on the part of the state, as well as a form of business for some entrepreneurs with the introduction of lotteries and casinos, respectively, and the novel conception of gambling as an economic activity. This new conception of gambling emerged as a result of the development of a new branch of mathematics: probability theory. Owing to new knowledge of probability, casinos were able to develop games with favorable odds—that is, games in which the benefit was always guaranteed for the house that organized the game. This benefit is naturally always at the players’ expense.

Gambling expanded considerably at the end of the 20th century in most countries where gambling was legal. In Spain, this expansion coincided with the legalization of casinos, gambling halls, and slot machines, which were also installed in bars and restaurants. What had once been a marginal activity controlled by “opaque” sectors (at least fiscally) developed into a business model in which players spent vast amounts of money in salons, casinos, machines, etc. The legalization of gambling allowed casinos, to become a key source of tax revenue for the state, as well as lucrative businesses [4].

During the closing decades of the 20th century, gambling was transformed “*from an activity that was tolerated, to a business to be encouraged*” [5]. Consequently, gambling legislation was “domesticated” and gambling policies were implemented with the primary aim of promoting business. With the social and economic expansion of gambling—an activity in which the casino always wins—the number of people who not only lost money but developed gambling disorders also increased. By the end of the 20th century, gambling had come to be regarded as a public health issue [6,7]. 

At the beginning of the 21st century, the internet exerted a major influence on gambling. Nowadays, all traditional gambling games have corresponding online versions, and several others have emerged, such as “hot bets”, which have been made possible by the development of information technologies and the unique environment created by the internet. Many countries are adapting their laws to permit and promote this online form of gambling activity that just a few years ago would have been deemed akin to science fiction [8]. 

Online gambling was legalized in Spain in 2011 to regulate an activity that had been (illegally) available on the internet since 2004. Finally, online gambling licenses (for casinos, bingo, slot machines, sports betting, and poker) began to be issued in June 2012. Since then, gambling offerings have greatly increased. Many new games have appeared, and thousands of venues have opened all over the country. The ruling that online gambling was not operating legally in Spain during that time was made by the Spanish Supreme Court in 2017 in response to a lawsuit filed by Codere against PokerStars.

Gambling is very profitable for companies because, regardless of the modality, the odds always favor whoever is managing the gambling activities [9]. Consequently, gambling is a highly lucrative activity for companies, who always profit in the long term, particularly when large amounts of money are exchanged. In addition, gambling involves a huge amount of money. Spending on gambling in Italy, Finland, and Spain exceeds 3% of the gross domestic product (GDP) [10]. The severity of gambling problems and the amount of money spent are positively correlated [11,12,13].

There are clear differences among the different types of gambling both in terms of spending and the gambling disorder they cause. Such differences are due to two factors: (a) structural characteristics of the different games [14], that is, the immediacy of the reward, the speed with which the bets take place, etc., and (b) environmental characteristics regulated by gambling policies [15], such as gambling offer, advertising, and promotion, gambling accessibility, etc.

In the case of Spain, the least addictive games (lotteries) are public (that is, managed by the State), while the most addictive ones (EGM, casinos, gambling halls, and online gambling) are private. The main companies are gambling multinational corporations. In the case of online gambling, many of them are based on tax havens to avoid paying corporate tax. The government of Spain in 2012 allowed it.

The evolution of gambling as an economic activity in some respects has shown a similar trend to other sectors, and some forms of gambling suffered during the 2008 economic crisis [16]. In Spain, for example, the relationship between GDP and spending on lotteries from 1995 to 2011 had a Pearson’s correlation coefficient (r^2^) of 0.98, indicating that the amount of money that people stake in lotteries aligns closely with what they can afford [17]. These results accord with the European Lotteries Report [18,19] and other studies [20], although studies carried out in some countries have concluded that spending on lotteries did not decrease during the financial crisis [21], or even increased [22,23,24].

No such relationship was observed in other contexts, such as slot machines and bingo, in a Spanish study, suggesting that, for these games, other variables are related to spending in addition to the gambler’s purchasing power [17]. People who suffer from gambling disorders tend to develop serious financial problems as a result of debts incurred due to gambling losses [13]. 

The ongoing COVID-19 pandemic caused by the SARS-CoV-2 infection has compelled governments to implement measures that significantly curtail offline gambling (lotteries, casinos, gambling halls, EGMs, etc.), particularly targeting the main variables: availability and accessibility. In the case of gambling, availability represents the supply of games in society, while accessibility denotes the ease with which such games can be played. The temporary closure of gambling halls and other betting venues, as well as capacity limitations, have resulted in reduced access to offline gambling and a decrease in gambling frequency during lockdown [25,26,27]. However, these restrictive measures had no impact on online gambling, which gamblers can engage in remotely using mobile devices, computers, or tablets, particularly in the home. Thus, the effects of the COVID-19 pandemic on gambling and problematic gambling have been diverse [28] and limited [29].

People who used gambling to cope with the crisis were found to be more vulnerable to gambling disorder than those who spent out of their savings (and did not depend on bank credit) or spent less on basic and non-basic needs [30]. Thus, social protection measures taken by governments are essential not only to overcome the economic crisis, but also to prevent gambling disorder. 

In summary, analyzing the relationship between economic crises and public health is important because one of the main consequences of such crises is increased poverty, which in turn is a major driver of deteriorating health. It has been found that people with low incomes are more vulnerable to gambling disorders [31,32,33,34,35].

Another reason why it is very important to analyze economic crises in the context of public health is to understand the measures that companies take to improve their profits by increasing consumption, which is the basis of the capitalist system. Such measures are not necessarily negative in the context of consumer goods, but in the case of the gambling sector, they may induce compulsive gambling, which is one of the most important aspects of gambling disorder [36,37].

Among this study’s key objectives is to analyze the impact of the two recent economic crises (i.e., the 2008 financial crisis and that precipitated by COVID-19) on gambling spending, taking into account the differences between the main gambling modes (i.e., offline and online).

## 2. Materials and Methods

### 2.1. Hypotheses

The two economic crises reduced spending on gambling, although differences will be evident among the different types of gambling in terms of the relationship between GDP and spending thereon.The measures taken to reduce access to gambling venues, to curb the spread of SARS-CoV-2, will cause a reduction in spending on offline games but will not affect spending on online gambling.

### 2.2. Analysis

Trends in spending on different types of games, the correlation between spending and GDP, and the existence (or not) of a regression function that relates GDP in relation to each type of game were analyzed. Regression analysis estimates the relationship between two variables using a mathematical function; more specifically, it allows the value of a variable to be predicted based on that of another known variable.

Gambling spending is the amount of money that people wager on gambling. That is, they are the gross income of gambling companies. It is a different concept from gross gambling revenue (GGR), which is what gambling companies do not return to players and would therefore be the net income of gambling companies. 

According to OECD, the gross domestic product, or GDP, is the standard measure of the value created by producing goods and services in a country during a certain period. In our study, GDP was the criterion used to determine the extent of the economic crisis. 

The different games are grouped into two categories: offline gambling (lotteries, slot, casino games) and online gambling (e-slot, e-roulette, e-bets, and e-poker). 

### 2.3. Procedure

We consulted Spanish databases comprising reliable records of spending on different gambling games. Comprehensive records of gambling spending in Spain date back to 1995, while data on online gambling have been collected since 2013 because companies were granted licenses to operate in Spain in mid-2012. 

Data were collected from the following public and private sources to create the databases used in this study:Directorate General for the Regulation of Gambling (DGOJ), Ministry of Consumer Affairs, Government of Spain, to obtain online gambling data and historic offline gambling data [38].National Institute of Statistics (INE), an autonomous institution of the Government of Spain, to obtain GDP data [39].Spanish Gaming Business Confederation (CEJUEGO), a business organization integrated within the State Confederation of Business Organizations (CEOE), which is the main organization representing large companies in Spain [40], to obtain recent offline gambling data.

## 3. Results

### 3.1. Global Spending on Traditional (Offline) Gambling and Online Gambling

Table 1 shows the main global economic data (gambling spending and GDP) during the period 1996–2020, which covers the two main crises (2008 and 2020). Figure 1 illustrates the Spaniards’ total spending on gambling in this period. A distinction is made between traditional offline games (lotteries, slot machines, casinos, etc.) and online gambling (sports betting, roulette, slots, etc.).

The period between 1996 and 2020 included three major milestones that affected gambling spending in Spain: (a) the 2008 financial crisis; (b) the legalization of online gambling in 2012, and (c) the COVID-19 crisis.

Regarding offline gambling, following sustained spending growth since 1996, a turning point occurred coincident with the 2008 financial crisis, and a decline in overall spending ensued. Spending recovered slowly from 2015 onward. The second turning point occurred in 2020, when a drastic reduction in spending, of 30.7%, was recorded in a single year owing to the COVID-19 pandemic.

By contrast, spending on online gambling has grown steadily since its legalization in 2012 (amid the economic crisis) but was unaffected by the 2008 financial crisis and the COVID-19 pandemic. Since its legalization, it has shown positive growth every year, with an average annual growth rate of 21.35% (range: 8.98–30.93%). Despite the crisis caused by the pandemic, the growth in 2020 relative to 2019 was 14.29%. Figure 2 illustrates the evolution of online gambling spending since it was legalized. 

In general terms, spending on online gambling has exhibited constant growth since it was legalized. The polynomial regression curve is described using the formula:y = 0.2164 × 2 − 869.54x + 873,468 (*R*^2^ = 0.98; *F* = 255.10; *p* < 0.001)

### 3.2. Spending on the Different Types of Gambling

#### 3.2.1. Evolution of Spending on Different Games between the 2008 Financial Crisis and the COVID-19 Pandemic

Data pertaining to the evolution of spending on different games are shown in Figure 3.

Regarding offline gambling, spending on lotteries, casinos, and slot machines fell during the crisis years. From 2014 onward, spending increased again, coinciding with the partial economic recovery. The 2020 COVID-19 pandemic caused by SARS-CoV-2 caused a reduction of approximately 29% in spending on gambling. The decrease in spending was less pronounced for lotteries than for slot machines and casino games. Spending on lotteries fell by 17.4% as a result of COVID-19, while the reduction in spending on casinos and slot machines was >40%.

Regarding online gambling, sustained spending growth has been seen since its legalization in 2012, in the cases of e-betting, e-slots, and e-casinos (roulette and blackjack). E-bets showed a decline in spending growth in 2018, with spending remaining stable at around 7000 million euros, while e-slots and e-casinos continued to grow in 2020 following the emergence of the COVID-19 pandemic. E-poker showed a decline in spending during the first few years of legalization, i.e., up to 2017, but showed growth thereafter that was highest in 2020.

The results of regression analyses for each of the online games are presented in Figure 4 and Figure 5. Data are billions of Euros.

Online spending on sports betting has exhibited constant growth since its legalization. The regression curve is described in a straight line using the formula:y = 0.8789x − 1767.3 (*R*^2^
*=* 0.91; *F =* 67.53; *p <* 0.001)

Spending on online casino gambling has exhibited constant growth since it was legalized. The regression curve is described in a straight line using the formula:y = 0.768x − 1545.1 (*R*^2^
*=* 0.98; *F =* 448.13; *p* < 0.001)

A regression curve predicting the temporal evolution of online poker could not be devised.

Spending on online slot machine games has exhibited constant growth since its legalization. The regression curve is a straight line described using the formula:y = 1.0134x −2041.7 (*R*^2^ = 0.996; *F* = 1107.43; *p* < 0.001)

#### 3.2.2. Relationship between Spending on Different Games and GDP

Table 2 details the spending on the main types of gambling, as well as the Spanish GDP during the period 1996–2020.

The results of regression analyses of GDP (X-axis) and spending on the different types of gambling (Y-axis) are presented in Figure 6, Figure 7, Figure 8 and Figure 9. Data are billions of Euros. 

Offline gambling

Regression analyses of GDP and lottery spending established a polynomial regression model:y = −9 × 10^−6^x^2^ + 0.0199x − 0.1728 (*R*^2^ = 0.80; *F* = 92.57; *p* < 0.001)

No relationship was identified between GDP and slot machine spending.

Regression analyses between GDP and lottery spending established a polynomial regression model:y = 5 × 10^−6^x^2^ − 0.0049x + 6.3075 (*R*^2^ = 0.69; *F* = 19.71; *p* < 0.001)

Offline gambling

No relationship was identified between GDP and e-Poker spending.

Regression analyses between GDP and online sports betting spending established a polynomial regression: y = −9 × 10^−5^x^2^ + 0.2262x − 136.81 (*R*^2^ = 0.73; *F* = 19.71; *p* < 0.001)

No relationship was found between GDP and eCasino spending.

No relationship was identified between GDP and online slot machine game spending.

## 4. Conclusions and Discussion

The COVID-19 pandemic has had a dramatic impact on the economy [41], and on human physical [21] and mental health [42,43]. Moreover, the measures that governments were obliged to implement to curb the spread of the virus drastically affected social and economic activities; gambling is no exception to this [44,45]. This article analyzed the effects of two global crises—the economic crisis caused by the financial scam of subprime mortgages, and the crisis precipitated by the spread of SARS-CoV-2—on gambling practices. This study demonstrates that, in terms of public health, the impact of economic crises on gambling spend largely depends on two key variables: (a) the addictive potential of the type of gambling itself (where less addictive games, such as lotteries, are more significantly affected by purchasing power) and (b) the impact of the social measures taken to prevent the spread of SARS-CoV-2, which restricted the availability and accessibility of gambling. Lotteries were less severely impacted by these measures than casinos and slot machines, while online gambling was not affected by either crisis. These results partially confirm the hypotheses proposed herein.

Regarding the first crisis (2008), the decline in Spaniards’ purchasing power—evaluated in terms of GDP—clearly impacted gamblers’ spending on offline games [17]. In fact, the 5 years that followed the financial crash were characterized by a significant decrease in spending, of 23.4%; this only began to recover in 2014, coinciding with a slight rebound in economic activity. 

However, analysis of the different gambling modalities revealed substantial differences among them. For example, lotteries showed a high positive correlation with GDP (r^2^ = 0.90) in the period 1996–2021. The data show a linear relationship between GDP and lottery spending. This result agrees with the 2011 European Lottery Report [18], which revealed that the compound annual growth rate of lotteries after the financial crisis, i.e., during 2009–2012, was negative [19]. Similarly, other studies found that the lottery showed the least decline after the stock market fall [20]. Nevertheless, it contrasts with that obtained by other studies, such as The Netherlands, in which lottery consumption appeared to be recession-proof because they showed vast and solid growth during economic expansions and recessions [21]. In the case of Iceland, there was also growth in spending on lotteries during the economic collapse in 2008, while such growth did not occur in the other types of gambling [23,24] 

These differences may be due, among other causes, because lotteries are extremely popular in Spain, with approximately 70% of the population participating annually [46]; however, lottery spending accounts for only 25% of all spending on offline games. This indicates that gamblers do not risk excessive amounts of money in lotteries compared to other games. This stands in contrast to the higher per capita spending on casinos and EGMs, since 65% of spending on offline gambling (casino games and EGMs) was accounted for by only 10% of the Spanish population. 

The significant correlation between lotteries and GDP in our study also indicates that Spaniards spend on lotteries only what they can afford. Thus, spending is lower during crises, despite the sales infrastructure (lottery sales offices) remaining the same. This may be attributed to lotteries’ lower potential for addiction than other modalities, such as EGMs, which have the greatest addictive potential [47,48]. 

A statistically significant, but much smaller, Pearson’s correlation coefficient was also found for casinos and GDP (r^2^ = 0.59); as such, we cannot draw the same conclusions as for the lottery. The number of players who participate in casino games is significantly lower than those who play lotteries, which indicates that they have a much higher per capita spending. Moreover, the percentage of pathological players is likely to be considerably higher in casinos and gambling halls than among those who play lotteries, and studies have shown that a sizable portion of casinos’ income is derived from pathological gamblers [49,50].

No relationship was observed between spending on slot machines and GDP. This may be due to the high percentage of pathological gamblers among EGM gamblers, as this game is particularly harmful to those who are undergoing treatment for gambling disorder [51]. In many cases, gambling disorder leads to financial ruin. In the context of slot machines, unlike lotteries, gambling behavior occurs independently of whether the gambler has financial resources. In fact, most pathological gamblers have grave financial problems. These results are partially in agreement with a study carried out in Italy showing that the relationship between the economic crisis and gambling spending varied among different types of gambling. While expenditure on so-called “luck” games (i.e., traditional lotteries, video lotteries, and slot machines) was positively correlated with unemployment, this was not the case for skilled gambling (horse racing, sports betting, and online games) [52]

Online gambling was legalized in Spain amid the 2008 economic crisis, during which GDP declined. Despite this, spending on online gambling has consistently grown for e-casino games (e-roulette, e-blackjack), e-slot, and e-sports betting. 

The progressive growth in online gambling offers provoked two distinct reactions from the traditional gambling companies that were operating in Spain until online gambling was legalized. First, the number of game rooms increased [40] in a bid to compete against the multinational gambling companies that had obtained a license to operate in Spain (e.g., Bwin, Pokerstars, 888, Unibet, Bet365, etc.). This was one of the factors that promoted growth in offline gambling spending during the recovery from the 2008 crisis. Second, the main Spanish companies that owned gambling rooms and EGMs created their business division for online gambling (e.g., Sportium from Cirsa, Codere online from Codere, Juegging from Orenes, Luckia from Egasa, etc.), causing online gaming to expand significantly. The increased availability and accessibility of gambling were conducive to both the recovery of offline gambling and continued growth in spending on online gambling. Consequently, the prevalence of addiction to online gambling increased considerably [53]. 

Online poker merits a special mention. Poker was the first online game available in Spain, at a time when companies still lacked the authorization to operate (the existing legislation at that time did not cover this novel activity). Poker games were provided by websites operating outside of Spain, and gamblers could participate from anywhere in the world. When online gambling was legalized in 2011, and licenses granted in 2012, gaming tables were limited to players whose computers’ internet provider (IP) addresses were located in Spain, largely due to fiscal control policies. This led to a decline in spending, as many professional players left Spain for other countries, such as the United Kingdom and Malta, where potentially more profitable tables were available. In 2018, however, the Spanish government unexpectedly permitted gamblers to play against players whose computers IP addresses were located in Spain, Italy, Portugal, or France. This resulted in increased spending on online poker, as Figure 6 illustrates. Spending on poker shows no relationship to GDP, suggesting that most spending on online poker is done by pathological gamblers and professional players.

The COVID-19 pandemic caused a completely unforeseen global health and economic crisis. In 2020, the majority of governments implemented measures restricting mobility and social contact to prevent the spread of SARS-CoV-2. These restrictions affected the economy drastically, causing a reduction in GDP while also affecting several social behaviors. However, not all activities have been equally impacted: while behaviors that entail physical proximity to people have been notably reduced, those based on the internet have not been substantially reduced; in fact, some activities, such as electronic commerce, have witnessed considerable growth.

In the case of gambling, physical proximity restrictions affect the two main variables underlying consumption and addiction discussed above: availability and accessibility. Availability refers to the gambling activities on offer, while accessibility refers to the ease with which a player can place bets. In this respect, differences between traditional (offline) gambling and online gambling are evident, since offline gambling takes place in public locations, such as gambling halls, casinos, and—in Spain—the interiors of bars and restaurants. Spain’s pubs and bars contain more than 170,000 slot machines, and no attempts have been made to control access thereto. By contrast, online gambling can take place in the privacy of the home—that is, in conditions that do not violate social distancing policies designed to curb the spread of COVID-19.

Offline gambling was thus limited in terms of availability with the closure of public gaming venues (gambling and bingo halls, casinos, bars, pubs, etc.) during the severest restrictions associated with the pandemic. Gambling halls and bars subsequently reopened, but with reduced capacity, leading to some degree of restricted access. Personal identification was also required, which further reduced accessibility. Consequently, a significant drop in gambling spending of approximately 40% was reported by gambling halls and casinos [40]. Other offline games not subject to such severe restrictions, or in which social distancing was irrelevant, such as lotteries, reported a smaller decrease in spending of approximately 17% [54].

Online gambling, which is practiced via smartphones, tablets, and computers, suffered no decline; on the contrary, spending on online gambling continued to rise, especially due to the sustained increase in online casino games (roulette, blackjack) and slot, which exhibited growth of 25.2% and 21.5%, respectively [38]. The regression analyses reveal a linear relationship with continuous growth, despite the decline in GDP in the early stages of legalization and 2020. In fact, no relationship was identified between GDP and spending on online gambling. Only a polynomial relationship with sports betting was evident, due to the fact that it was the online game most affected by anti-COVID measures (e.g., the suspension of matches and the recent ban on sports betting advertising in Spain). Sports betting appears to have ceased growing in 2018, although during 2020 this stagnation was mainly attributable to the suspension of many sports competitions worldwide during April and May, with betting showing a concomitant quarterly decline of approximately 50%. The return of sports competitions in turn saw the return of pre-pandemic betting levels; these were even exceeded during the Tokyo Olympics. Online poker showed the greatest increase during the COVID-19 crisis, with 28.4% more spending reported than in 2019.

In the case of slot online games, the regression analysis yielded a straight regression line and r^2^ of 0.996, indicating constant growth since its introduction, despite the crises. The exponential growth of e-slot options, together with the addictive potential of this type of game, suggest that e-slots will become the main problematic form of gambling in the near future.

This study had several limitations. First, as it was a correlational study, no conclusions regarding causality can be drawn. Nevertheless, the effects of gambling availability and accessibility on the likelihood of addiction are widely accepted by the scientific community. As such, the evident decrease in spending on offline games in the wake of the COVID-19 pandemic is highly relevant. Second, the data in this study do not reflect spending on illegal gambling, which is nonetheless relevant to online gambling. According to the European Gaming and Betting Association (EGBA), the offshore online gambling market accounted for 20% of all spending on this type of gambling in 2020. Although the extent of illegal online gambling remains unknown, the legalization of online gambling has led to a major increase in gambling spending by the general population. In the case of offline gambling, unlike some Latin American countries, there is strict control of most slot machines (offline games) in Spain, and the government collects taxes for EGMs (around 1 billion euros per year). Finally, it is difficult to determine whether the growth of online gambling has led to a decline in traditional gambling through a “cannibalistic” mechanism. To better understand this, other variables would have to be taken into account, such as whether there has been a movement of gamblers towards online gambling or if, on the contrary, they have taken over a new “market niche”. That’s what young people are for gambling companies.

In summary, the effects of the two recent global crises on gambling were markedly different. Although both events reduced purchasing power, the difference between the financial crisis of 2008 and that precipitated by the COVID-19 pandemic was that, in the latter, in addition to the reduction in purchasing power, the availability and accessibility of offline games were reduced as a result of measures leading to confinement and social isolation. This caused a considerable reduction in offline gambling. Online gambling platforms, however, allow participation without the need to visit public locations or come into contact with other people; the growth in spending thereon is related to the gambling activities offered and ease of playing. Online gambling also has the potential to be highly addictive, which has contributed to its growth; as of this writing, no deceleration in spending growth is anticipated.

The real problem is that gambling companies’ income comes directly -and exclusively- from gamblers’ losses. Moreover, the benefits are greater the more players gamble, because the mathematical expectation of all games always favors the company [9]. Based on the above discussion, gambling policies must be reconsidered to set moral limits for the market [55] reducing the availability and accessibility of offline and online games. Public authorities should prioritize the prevention of health problems over economic profits for gambling companies [56].

## Figures and Tables

**Figure 1 ijerph-20-02909-f001:**
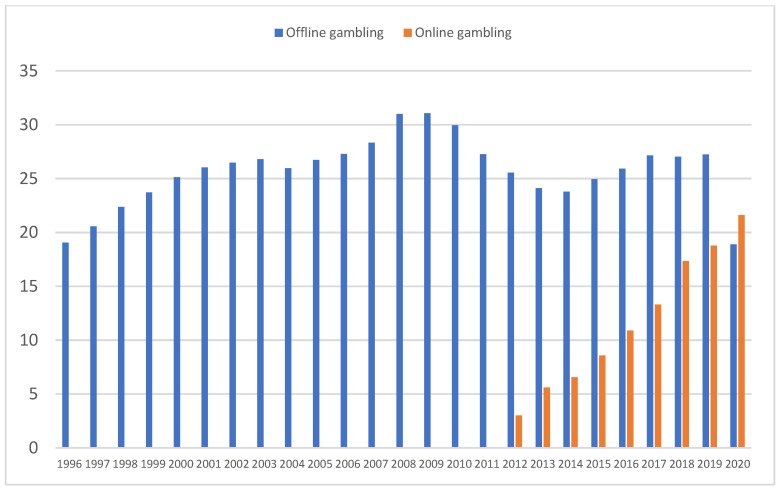
Gambling spending in Spain (1996–2020) (data are billions of Euros).

**Figure 2 ijerph-20-02909-f002:**
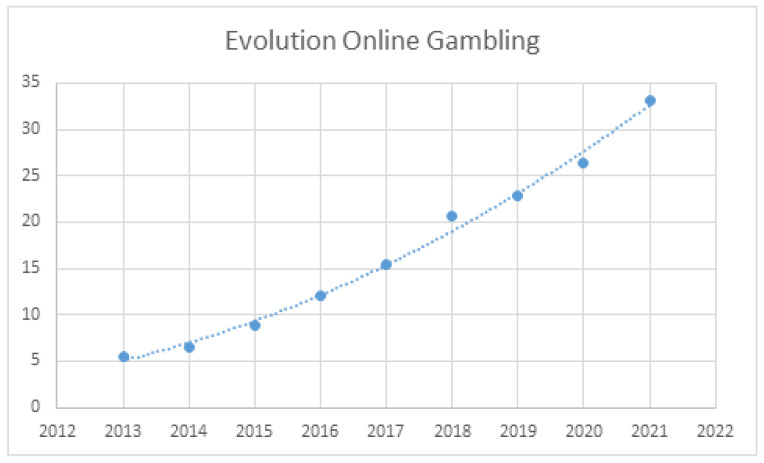
Evolution of online gambling (2013–2021) in Spain (data are billions of Euros).

**Figure 3 ijerph-20-02909-f003:**
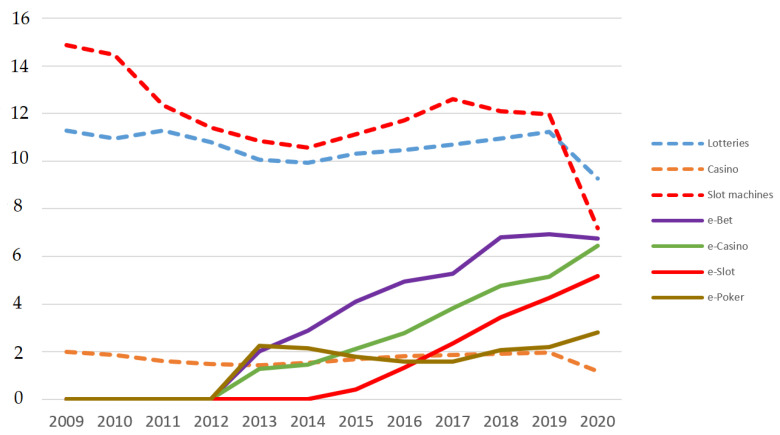
Spending on each type of gambling game (offline and online) (data are billions of Euros).

**Figure 4 ijerph-20-02909-f004:**
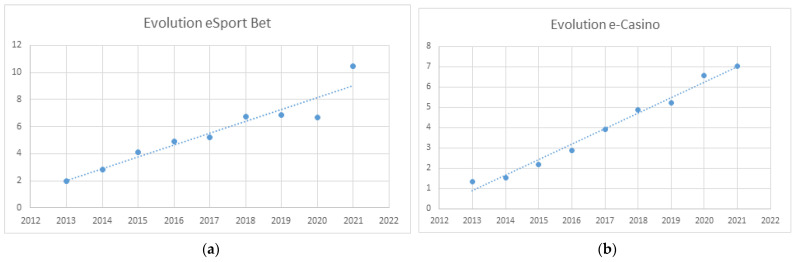
(**a**) Evolution of online sports betting spending; (**b**) Evolution of online casino game (e-roulette and e-Blackjack) spending.

**Figure 5 ijerph-20-02909-f005:**
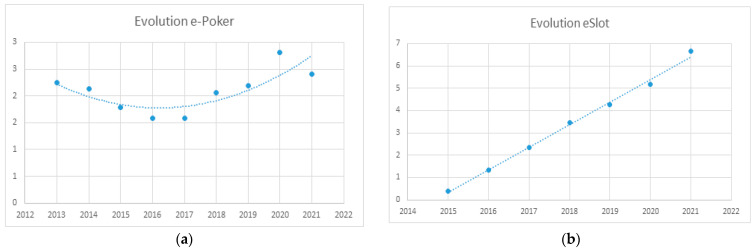
(**a**) Evolution of online poker; (**b**) Evolution of online slot-game spending.

**Figure 6 ijerph-20-02909-f006:**
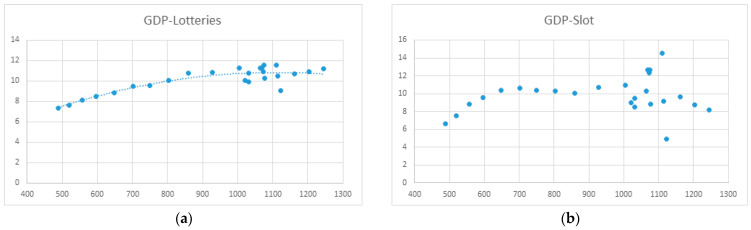
(**a**) Regression function between GDP and lottery spending; (**b**) Regression function between GDP and EGM spending (slot).

**Figure 7 ijerph-20-02909-f007:**
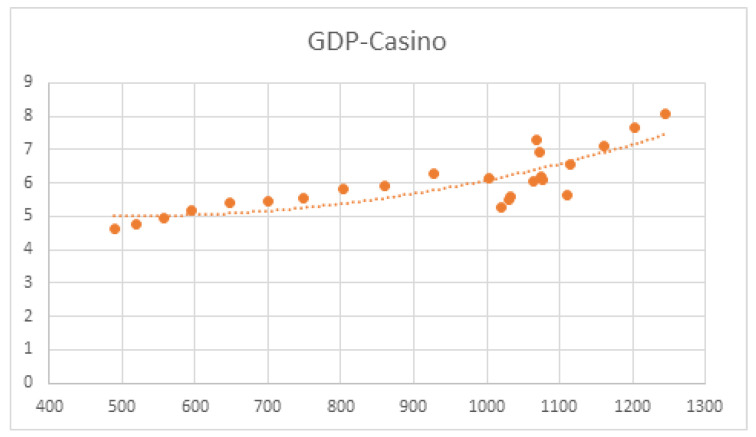
Regression function between GDP and casino spending.

**Figure 8 ijerph-20-02909-f008:**
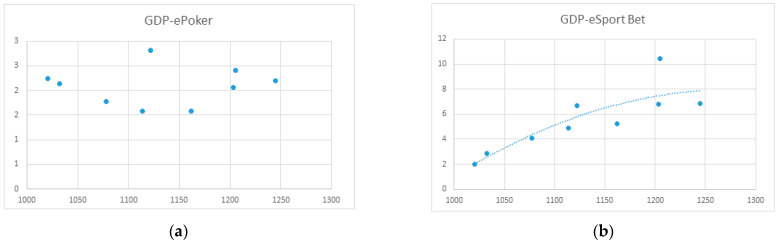
(**a**) Regression function between GDP and online poker; (**b**) Regression function between GDP and online sports betting.

**Figure 9 ijerph-20-02909-f009:**
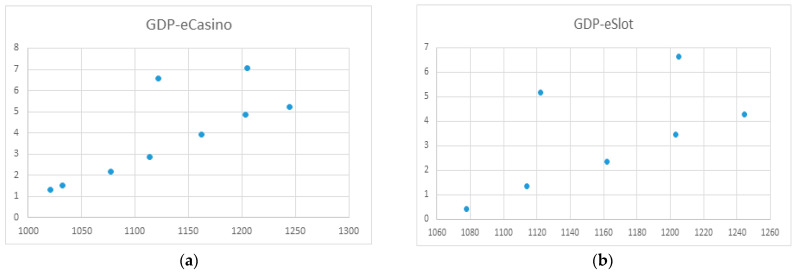
(**a**) Regression function between GDP and online casino spending; (**b**) Regression function between GDP and online slot spending.

**Table 1 ijerph-20-02909-t001:** Gambling spending (billions of Euros).

	**1996**	**1997**	**1998**	**1999**	**2000**	**2001**	**2002**	**2003**	**2004**	**2005**	**2006**	**2007**	**2008**
Offline	19.05	20.55	22.37	23.71	25.13	26.04	26.47	26.80	25.96	26.73	27.29	28.33	30.99
Online	-	-	-	-	-	-	-	-	-	-	-	-	-
Total	19.05	20.55	22.37	23.71	25.13	26.04	26.47	26.80	25.96	26.73	27.29	28.33	30.99
GDP	489.2	519.3	556.0	595.7	647.9	701.0	749.6	802.3	859.4	927.4	1003.8	1075.5	1109.5
	**2009**	**2010**	**2011**	**2012**	**2013**	**2014**	**2015**	**2016**	**2017**	**2018**	**2019**	**2020**	
Offline	31.06	29.96	27.25	25.55	24.11	23.79	24.94	25.91	27.15	27.02	27.24	18.88	
Online	-	-	-	3.00	5.56	6.51	8.87	12.00	15.43	20.58	22.85	26.40	
Total	31.06	29.96	27.25	28.55	29.70	30.35	33.50	36.80	40.45	44.35	46.02	40.48	
GDP	1069.3	1072.7	1063.8	1031.1	1020.3	1032.2	1077.6	1113.8	1161.9	1203.3	1244.4	1121.9	

**Table 2 ijerph-20-02909-t002:** GDP and gambling spending (billions of Euros).

	GDP	Lotteries *	Slot Machines *	Casino *	e-Poker	e-Sport Bet	e-Slot **	e-Casino	Online Gambling
1996	489.2	7.4	6.6	4.6	-	-	-	-	-
1997	519.3	7.6	7.5	4.8	-	-	-	-	-
1998	556.0	8.1	8.8	4.9	-	-	-	-	-
1999	595.7	8.5	9.6	5.2	-	-	-	-	-
2000	647.9	8.8	10.4	5.4	-	-	-	-	-
2001	701.0	9.5	10.6	5.4	-	-	-	-	-
2002	749.6	9.6	10.4	5.6	-	-	-	-	-
2003	802.3	10.1	10.3	5.8	-	-	-	-	-
2004	859.4	10.8	10.1	5.9	-	-	-	-	-
2005	927.4	10.8	10.7	6.3	-	-	-	-	-
2006	1003.8	11.3	10.9	6.2	-	-	-	-	-
2007	1075.5	11.6	12.6	6.2	-	-	-	-	-
2008	1109.5	11.6	14.5	5.7	-	-	-	-	-
2009	1069.3	11.3	12.7	7.3	-	-	-	-	-
2010	1072.7	11.0	12.3	6.9	-	-	-	-	-
2011	1063.8	11.3	10.3	6.0	-	-	-	-	-
2012	1031.1	10.8	9.5	5.5	-	-	-	-	-
2013	1020.3	10.0	9.0	5.2	2.2	2.0	-	1.3	5.6
2014	1032.2	9.9	8.5	5.6	2.1	2.9	-	1.5	6.5
2015	1077.6	10.3	8.8	6.1	1.8	4.1	0.4	2.2	8.9
2016	1113.8	10.5	9.1	6.6	1.6	4.9	1.3	2.9	12.0
2017	1161.9	10.7	9.6	7.1	1.6	5.2	2.3	3.9	15.4
2018	1203.3	11.0	8.7	7.7	2.1	6.8	3.4	4.9	20.6
2019	1244.4	11.2	8.2	8.1	2.2	6.9	4.3	5.2	22.8
2020	1121.9	9.1	4.9	4.4	2.8	6.7	5.2	6.6	26.4
2021	1205.1	-	-	-	2.4	10.5	6.7	7.1	33.2

* No official data for spending on offline gambling are available as yet. ** Online slot gambling was legalized in July 2015.

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
