# Peer review of "Crisis, What Crisis? The Effect of Economic Crises on Spending on Online and Offline Gambling in Spain: Implications for Preventing Gambling Disorder"

_ijerph, 2023, doi:10.3390/ijerph20042909_

Round 1

Reviewer 1 Report

I  enjoyed reading this manuscript which provides an interesting and important insight on the relationship between social event and gambling. Very minor suggestions could be adapted for improvement. 

1. it would be good to highlight to contribution of the current study findings for practice/research. e.g., why we need to investigate the impact of two economic crisis on gambling in the introduction. 

2. have ethics considerations been taken prior to the study? 

3. I suspect that the impact of Covid on individual finance status may not be immediately shown at the individual level, since social benefits/financial supports have been provided by the governments in many countries (not sure if this is the case in Spain) and personal daily spending somehow becomes less, e.g., clothes, sports, travel, dine out. If this is the case, would this affect the present findings. I would be interested to know the author's thoughts on this.   

Author Response

Reviewer 1

I  enjoyed reading this manuscript which provides an interesting and important insight on the relationship between social event and gambling. Very minor suggestions could be adapted for improvement. 

  1. it would be good to highlight to contribution of the current study findings for practice/research. e.g., why we need to investigate the impact of two economic crisis on gambling in the introduction. 
  2. have ethics considerations been taken prior to the study? 
  3. I suspect that the impact of Covid on individual finance status may not be immediately shown at the individual level, since social benefits/financial supports have been provided by the governments in many countries (not sure if this is the case in Spain) and personal daily spending somehow becomes less, e.g., clothes, sports, travel, dine out. If this is the case, would this affect the present findings. I would be interested to know the author's thoughts on this.   

Responses to Reviewer 1

  1. Analyzing the relationship between economic crises and public health is important because one of the main consequences of such crises is increased poverty, which in turn is a major driver of deteriorating health. It has been found that people with low incomes are more vulnerable to gambling disorder. Another reason why it is very important to analyze economic crises in the context of public health is to understand the measures that companies take to improve their profits by increasing consumption, which is the basis of the capitalist system. Such measures are not necessarily negative in the context of consumer goods, but in the case of the gambling sector they may induce compulsive gambling, which is one of the most important aspects of gambling disorder.

  1. The data for this study come from public databases made by public institutions (Spanish Ministry of Consumption, Spanish National Institute of Statistics), as well as surveys of the gambling industry (State Confederation of Gambling). This study is in accordance with the ethical standards of the Spanish government and with the 1964 Helsinki Declaration and its later amendments. All data are anonymous and are in accordance with Law 3/2018, on the protection of personal data and guarantee of digital rights.

  1. In general terms, taking into account the data of the entire population, purchasing power has a direct relationship with gambling spending. But there are several important issues to keep in mind. First, the type of game: the most addictive games cause much higher spending per capita, and they are the ones that cause personal ruin. Very few people go bankrupt playing lotteries. Most pathological gamblers are broke or spend a lot of money (more than they can afford) usually playing slot machines, casino games or online gambling.

Second, the measures taken to overcome the crisis affect gambling spending only if they reduce the availability of games. This has been seen by comparing the two crises. In the first crisis, no measures were taken to reduce the availability of the games. So, the crisis had no effect on the most addictive games; only in lotteries. In the second crisis, spending was significantly reduced in the games in which the availability was reduced (even in the most addictive games). In the case of online gambling, where there were no restrictions, the crisis had no effect.

For this reason we have added to the title of the article: "Crisis, what crisis?". It is a pun to indicate that the economic crises themselves only affects some games, as well as to point out that the differences on public measures taken for the governments to overcome the crises are very relevant to prevent the appearance of gambling disorder.

Reviewer 2 Report

This paper presents the effects of the two economic crises (the 2008 financial crisis and COVID-19) on gambling spending in Spain, taking into account the differences between the main gambling modes (offline and online). One novelty aspect of this paper is to look at the impacts of two economic crises on gambling spending in one paper. Overall, in the gambling research field and also from the gambling harm prevention viewpoint, there is still a need to understand the effects of such crises on gambling in different countries and within the different legal frameworks, although many papers have already been published (e.g. Iceland, Italy, US). However, the manuscript in its current format is not yet scientifically strong enough to warrant publication without significant revisions. Below, I have chosen to point out some concerns regarding the paper.

First, I think that references to previous work in the gambling field is largely lacking and/or are at least, somewhat selective. This concerns Introduction, but also Discussion, in particular, where I as a reader, would expect to see how the current findings accord with other studies conducted in the field in other countries such as in Italy (Capacci, S., Randon, E., & Scorcu, A. E. (2017). Are consumers more willing to invest in luck during recessions? Italian Economic Journal, 3, 25–38.), in Iceland (Olason, D. T., Hayer, T., Meyer, G., & Brosowski, T. (2017) Economic recession affects gambling participation but not problematic gambling: Results from a population-based follow-up study. Frontiers in Psychology, 8, 1247) etc.

Second, I am confused about the terms the authors have used throughout the paper: “gambling addiction”/ “gambling disorder” and “pathological gambling”. There is no clear definitions of these terms or how gambling addiction was defined. The authors also state in their study title :“Implications for gambling addiction”. This is problematic. Hypotheses do not say anything about it. No data about the changes in population-based gambling prevalence over time in Spain is not presented.

Third, the manner in which the authors report their material and methods is inadequate from the scientific viewpoint. For example, definitions regarding gambling spending, GDP, and gambling modes are missing. Furthermore, there are no description of the regression analyses applied in the study. 

Introduction: The author state that “In the case of gambling, availability represents the supply of games in society, while accessibility denotes the ease with which such games can be played. High levels of both variables increase citizens' exposure to gambling and its effects”. References are missing.

row 76: …..the damage they cause (gambling disorder). It looks for me that the authors think to believe that the damage gambling may cause is related only to the most severe form of problem gambling. However, gambling harm occurs at all (gambling frequency and individual, community, societal) level and are not solely restricted to problem gamblers. 

The results of this study also bring up the question about how the socioeconomic status of the Spanish population has changed over time. It has been demonstrated in the gambling literature, for example, that unemployment and low socioeconomic status is associated with higher gambling expenditure.

Author Response

This paper presents the effects of the two economic crises (the 2008 financial crisis and COVID-19) on gambling spending in Spain, taking into account the differences between the main gambling modes (offline and online). One novelty aspect of this paper is to look at the impacts of two economic crises on gambling spending in one paper. Overall, in the gambling research field and also from the gambling harm prevention viewpoint, there is still a need to understand the effects of such crises on gambling in different countries and within the different legal frameworks, although many papers have already been published (e.g. Iceland, Italy, US). However, the manuscript in its current format is not yet scientifically strong enough to warrant publication without significant revisions. Below, I have chosen to point out some concerns regarding the paper.

  1. First, I think that references to previous work in the gambling field is largely lacking and/or are at least, somewhat selective. This concerns Introduction, but also Discussion, in particular, where I as a reader, would expect to see how the current findings accord with other studies conducted in the field in other countries such as in Italy (Capacci, S., Randon, E., & Scorcu, A. E. (2017). Are consumers more willing to invest in luck during recessions? Italian Economic Journal, 3, 25–38.), in Iceland (Olason, D. T., Hayer, T., Meyer, G., & Brosowski, T. (2017) Economic recession affects gambling participation but not problematic gambling: Results from a population-based follow-up study. Frontiers in Psychology, 8, 1247) etc.

23 new references have been added in the sections dedicated to the introduction and discussion. Other investigations that studied gambling in some of the crises (both the 2007-2008 financial crisis and the one caused by COVID-19) in countries such as Italy, Iceland, Denmark, Greece, Sweden, etc., have also been described.

All changes made are in red in the text so they can be more easily identified by Reviewer.

  1. Second, I am confused about the terms the authors have used throughout the paper: “gambling addiction”/ “gambling disorder” and “pathological gambling”. There is no clear definitions of these terms or how gambling addiction was defined. The authors also state in their study title :“Implications for gambling addiction”. This is problematic. Hypotheses do not say anything about it. No data about the changes in population-based gambling prevalence over time in Spain is not presented.

Gambling disorder is a mental illness classified in the category of addictive disorders. That is, gambling disorder is an addictive disorder and, therefore, they are synonymous terms. This is considered by both the American Psychiatric Association (DSM-5) and the WHO (ICD-11). This classification is relatively recent (DSM-5 in 2013; ICD-11 in 2018). However, to avoid confusion, the term gambling addiction has been removed and replaced by gambling disorder.

  1. Third, the manner in which the authors report their material and methods is inadequate from the scientific viewpoint. For example, definitions regarding gambling spending, GDP, and gambling modes are missing. Furthermore, there are no description of the regression analyses applied in the study. 

Corrections suggested by the reviewer have been made:

Gambling spending is the amount of money that people wager on gambling. That is, they are the gross income of gambling companies. It is a different concept from gross gambling revenue (GGR), which is what gambling companies do not return to players and would therefore be the net income of gambling companies.

According OECD, the gross domestic product, or GDP, is the standard measure of the value created by producing goods and services in a country during a certain period. In our study, GDP was the criterion used to determine the extent of the economic crisis.

Regression analysis estimates the relationship between two variables using a mathematical function; mores specifically, it allows the value of a variable to be predicted based on that of another known variable.

The different games are grouped into two categories: offline gambling (lotteries, slot, casino games) and online gambling (e-slot, e-roulette, e-bets, and e-poker).

  1. Introduction: The author state that “In the case of gambling, availability represents the supply of games in society, while accessibility denotes the ease with which such games can be played. High levels of both variables increase citizens' exposure to gambling and its effects”. References are missing.

The paragraph has been written more correctly and references have been added

The ongoing COVID 19 pandemic caused by the SARS-Cov-2 infection has compelled governments to implement measures that significantly curtail offline gambling (lotteries, casino, gambling halls, EGMs, etc.), particularly targeting the main variables: availability and accessibility. In the case of gambling, availability represents the supply of games in society, while accessibility denotes the ease with which such games can be played. The temporary closure of gambling halls and other betting venues, as well as capacity limitations, have resulted in reduced access to offline gambling and a decrease in gambling frequency during lockdown [25-27].

  1. row 76: …..the damage they cause (gambling disorder). It looks for me that the authors think to believe that the damage gambling may cause is related only to the most severe form of problem gambling. However, gambling harm occurs at all (gambling frequency and individual, community, societal) level and are not solely restricted to problem gamblers. 

Yes, of course. Although gambling disorder is the main problem caused by gambling in terms of public health (in fact, it is one of the main causes of suicide), it is true that gambling results in many other problems. So, that there is no misunderstanding, the term damage has been removed.

  1. The results of this study also bring up the question about how the socioeconomic status of the Spanish population has changed over time. It has been demonstrated in the gambling literature, for example, that unemployment and low socioeconomic status is associated with higher gambling expenditure.

In this study, the data on the socioeconomic level of the gamblers have not been analysed, but rather the macroeconomic figures. However, studies that link poverty with gambling-related problems have been added in the Introduction section.

People who used gambling to cope with the crisis were found to be more vulnerable to gambling disorder than those who spent out of their own savings (and did not depend on bank credit) or spent less on basic and non-basic needs [30]. Thus, social protection measures taken by governments are essential not only to overcome the economic crisis, but also to prevent gambling disorder.  

Round 2

Reviewer 2 Report

The updated mansucript is much improved. The author responded to my comments and concerns at adequate level and revised the manuscript accordingly. I have no further comments.